# New Genomic Signals Underlying the Emergence of Human Proto-Genes

**DOI:** 10.3390/genes13020284

**Published:** 2022-01-31

**Authors:** Anna Grandchamp, Katrin Berk, Elias Dohmen, Erich Bornberg-Bauer

**Affiliations:** 1Institute for Evolution and Biodiversity, Westfälische Wilhelms University, Hüfferstrasse 1, 48149 Münster, Germany; k.berk@wwu.de (K.B.); e.dohmen@wwu.de (E.D.); ebb.admin@uni-muenster.de (E.B.-B.); 2Department of Protein Evolution, Max-Planck-Institute for Biology, 72076 Tübingen, Germany

**Keywords:** human proto-genes, introns, regulatory motifs, 5′ UTRs, protein domains

## Abstract

*De novo* genes are novel genes which emerge from non-coding DNA. Until now, little is known about *de novo* genes’ properties, correlated to their age and mechanisms of emergence. In this study, we investigate four related properties: introns, upstream regulatory motifs, 5′ Untranslated regions (UTRs) and protein domains, in 23,135 human proto-genes. We found that proto-genes contain introns, whose number and position correlates with the genomic position of proto-gene emergence. The origin of these introns is debated, as our results suggest that 41% of proto-genes might have captured existing introns, and 13.7% of them do not splice the ORF. We show that proto-genes which emerged via overprinting tend to be more enriched in core promotor motifs, while intergenic and intronic genes are more enriched in enhancers, even if the TATA motif is most commonly found upstream in these genes. Intergenic and intronic 5′ UTRs of proto-genes have a lower potential to stabilise mRNA structures than exonic proto-genes and established human genes. Finally, we confirm that proteins expressed by proto-genes gain new putative domains with age. Overall, we find that regulatory motifs inducing transcription and translation of previously non-coding sequences may facilitate proto-gene emergence. Our study demonstrates that introns, 5′ UTRs, and domains have specific properties in proto-genes. We also emphasize that the genomic positions of de novo genes strongly impacts these properties.

## 1. Introduction

*De novo* gene birth is defined as the emergence of a new gene from previously non-coding sequences [1,2,3,4]. Over the past 15 years, an increasing number of studies investigated *de novo* gene birth and its role in genome evolution [3,5,6,7]. The emergence of new genes depends mainly on two processes: the occurrence of a transcription event in a previously untranscribed DNA region, and the emergence of an open reading frame (ORF) encoding for a protein [6]. Putative *de novo* genes which appear via the emergence of an ORF and a transcription event are called proto-genes [8]. The mechanisms of proto-gene emergence differ depending on the genomic location in which the new gene emerges (Appendix A).

Among these mechanisms, overprinting is characterised by the overlap of a new gene with pre-existing exons [9,10]. If the overlap is in frame with the pre-existing exon, the resulting protein will be shorter or longer than the original one. If the overlap is not in frame, or on the complementary DNA strand, a completely new protein can emerge [11,12]. Overprinting can lead to other events. For example, *de novo* extension of an exonic open reading frame (ORF) can give rise to the overlap of distant genes. This process affects 5–14% of vertebrate genes [13].

Exonisation characterises the emergence of a new gene which overlaps with an intron [14,15]. The minimum requirement for such an event to occur is the retention of an intron as an exonic sequence, and the presence of an ORF inside. Proto-genes can also emerge “from scratch” in intergenic regions. This occurs when a transcript emerges from a previously untranscribed intergenic region, which also gains an ORF [16,17]. Emergences “from scratch” are difficult to explain, as all constitutive elements of a gene need to be present simultaneously. Since all sequence signals which are not constantly under purifying selection are likely to be lost again, their co-occurrence may be a consequence of their emergence in close temporal proximity. Accordingly, such an emergence must be considered very unlikely and certainly deserves more detailed investigations. Two additional mechanisms have been proposed to account for proto-gene emergence: an ORF can emerge in a long non-coding RNA (lncRNA), or a pseudogene can regain transcription [2]. However, we will focus on the first three mechanisms in this manuscript: overprinting, exonisation and emergence “from scratch”.

So far, several *de novo* genes have been shown to provide important functionality, for example, playing a role in reproduction [18,19], adaptation to new ecological niches (e.g.-antifreeze proteins enabling fish to survive in arctic water [20]), speciation events, etc. [21,22,23]. A major limitation in studying *de novo* genes is the difficulty in ascertaining that a proto-gene arose *de novo*, and did not emerge through rapid evolution or transposition of a previously existing gene. To avoid false positives in *de novo* gene identification, which is mainly due to divergence of homologues, detection procedures have been constantly refined [24,25]. In essence, to accomplish a high degree of selectivity, rigorous methods such as synteny and similarity profile searches need to be applied [5,26,27]. *De novo* proteins have been proposed to contain a high level of intrinsic structural disorder and lack a stable three-dimensional structure [28], even if that result has been challenged by [19], who did not observe distinguishable properties between *de novo* gene structure and randomly selected sequences. By studying viruses, [10] it was demonstrated that *de novo* genes which emerged via overprinting may have a different codon usage than established genes. Some *de novo* genes preferentially emerge next to promotors in GC-rich intergenic regions and can make use of bidirectional promotors [29]. *De novo* genes in plants have been shown to be more prone to emerge when they are in a higher epigenetic methylation state [30]. Upstream region of *de novo* genes have also been shown to exhibit an enrichment in transcription factors [23]. The expression of young genes is lower than that of established genes, and more tissue-specific [7,8]. Based on an evolutionary perspective, the older *de novo* genes become, the more constitutively expressed they are [31], and the more selected for [32]. However, the above-mentioned properties are not sufficient to distinguish proto-genes from new genes which have emerged through other mechanisms. Investigating new features of proto-genes may allow such a distinction, and improve the basic knowledge of *de novo* genes.

In this study, we aim to characterise properties of proto-genes, which have, to the best of our knowledge, not been quantitatively explored before, in order to improve their detection and description. Secondly, we aim to understand if the three mechanisms of emergence (“overprinting”, “exonisation”, or “from scratch”), as well as the approximate age of emergence, influence the properties of the proto-genes. We investigated four properties by studying 27,550 proto-genes from the human lineage published previously [33]: (1) the presence of introns, (2) the enrichment and nature of motifs specific for transcription, (3) the properties of 5′ untranslated region (UTR), and (4) the presence of protein domains.

We first studied the presence of introns (1). Introns are major constituents in eukaryote genomes, as they allow for alternative splicing, and therefore considerably increase the number of encoded proteins compared to the total number of genes [34]. Introns can be spliced by RNA catalysis, by proteins, or by the spliceosome [35,36,37]. In extant eukaryotes, the emergence of introns in genes can be described by different scenarios [38]: intron transposition, transposon insertion, tandem duplication, intron gain during double strand break repair, insertion of a group2 intron, intron transfer, and intronisation. Most proto-genes are thought to not contain introns [5,39] However, this assumption is based on small datasets. In this present manuscript, we examine whether proto-genes contain introns, and if intron size and number are correlated to the age and the mechanism of emergence of the proto-gene.

Secondly, we investigate the enrichment and nature of DNA motifs specific for the transcription of proto-genes (2). The mechanism of gene transcription is well known and includes complex binding of transcription factors in eukaryotes [40]. The binding of transcription factors is facilitated by specific DNA motifs in the promotor and in enhancer regions. The overall transcription is known to be less constitutive in proto-genes compared to older established genes [3]. However, little is known about the presence of motifs which allow recruitment of the transcription machinery.

Third, we studied 5′ UTRs (3) of proto-genes. 5′ UTRs are essential elements of the translation machinery of a gene [41,42]. In eukaryotes, the 5′ UTR, which is located upstream of the start codon, contains the Kozac sequence [43] which is recognised by the ribosome, supported by multiple factors. Furthermore, UTRs play a role in some biological processes, including modulating of mRNA transport out of the nucleus, and subcellular localisation [44]. 5′ UTRs are GC-rich in eukaryotes, which support stable secondary structure elements, playing a role in the transcription regulation [45,46]. Moreover, it has been proven that mutations in UTRs can deregulate transcription. Additionally, some diseases are caused by mutation in UTRs [47]. To our knowledge, 5′ UTRs have never been studied in proto-genes. Considering the proposed importance of 5′ UTRs for translation, it would be an important first step to investigate if 5′ UTRs present some intermediate characteristics in proto-genes in order to better understand how proto-genes become translated into proteins. Accessing the structure of 5’UTRs of proto-genes would also highlight any specificity that could differentiate proto-genes from established genes or from non-coding sequences. In the present manuscript, we investigate the folding potential of the 5′ UTR in proto-genes and compare this to the folding potential of established human genes.

As a last step, we studied protein domains (4). Domains are structural or functional units of proteins which fold in an independent way and have a characteristic pattern of hydrophobic and polar residues [48,49]. Most domains are shared by many proteins and organisms [50], and rearrangements of their order are frequent [51,52]. Proto-genes are thought not to contain any known domains [25]. However, as domains rearrange through different molecular mechanisms, proto-genes might acquire existing domains, e.g., by fusion with another long existing gene coding for such a domain. In the present study, we investigate whether proto-genes contain known domains, and if there is a correlation between the presence of a known domain in proto-genes and their age and mechanism of emergence. We also investigate if putative proto-proteins (the protein sequences potentially encoded by a proto-gene) have a potential to form novel domains. This would be conceivable, considering that a recent study found that, presumably via occasional read through [53], it is possible that a reading frame becomes extended [54] and eventually some of the extended fragments become stabilised by forming hydrophic interactions [55].

We used the four properties (presence of introns, enrichment and nature of motifs specific for transcription, properties of 5′ UTR, and presence of domains,) to compare proto-genes which emerged *de novo* to established human genes, in order to determine if the mentioned properties allow a distinction between the two gene classes. Moreover, we classified proto-genes according to their genomic position which depict their mechanism of emergence (“overprinting”, “exonisation”, or “from scratch”), as well as by their evolutionary age.

## 2. Materials and Methods

### 2.1. Datasets

The human genome GRCh38 and its corresponding GTF file were downloaded from Ensembl [56]. The 27,491 putative human *de novo* transcribed ORFs, nucleotide sequences and corresponding protein sequences, were provided by [33]. ORFs whose coding sequence, provided by [33] (Appendix A), did not entirely match their corresponding genomic sequence were removed from analyses, as these contain potential mapping artifacts. This gave rise to a set of 23,135 putative human *de novo* transcribed ORFs. The spliced transcripts containing *de novo* transcribed ORFs as well as the genomic positions and exon-intron structure of the corresponding unspliced transcripts (GTF file) were provided by [33]. Following the definition from [8], we here denote as “proto-gene” any *de novo* transcribed ORF (provided by [33]), with its entire genic structure: the UTRS, and the intron exons of the entire transcript on which the ORF is located” (Figure 1).

Proteins encoded by putative proto-genes are mentioned as proto-proteins. The genomic contexts in which the proto-genes emerged are described in this study as “genomic position”. The three genomic positions directly refer to the mechanism of ORF emergence: “exonic” ORFs = overlap with an annotated exon (overprinting), “intronic” ORFs = overlap with an intron (exonisation), and “intergenic” ORFs = emerged inside a non-coding region and therefore from scratch. Ages of the proto-genes were annotated as following: I0 = specific to humans (*Homo sapiens)*; I1 = emergence at 6.65 Mya, shared between humans, bonobo (*Pan paniscus*), and chimpanzee (*Pan troglodytes*); I2 = 9.06 Mya (I1 + gorilla (*Gorilla gorilla*)); I3 = 15.76 Mya (I2 + orangutan (*Pongo pygmaeus*)); I4 = 29.44 Mya (I3 + rhesus macaque (*Macaca mulatta*)); I5 = 90 Mya (I4 + mouse (*Mus musculus*) as outgroup)). Proto-genes from age class I5 are not primate-restricted, but they are considered as outgroup proto-genes in the present study.

### 2.2. Intron Search

The exon/intron structure of the unspliced transcript of all proto-genes was extracted from the GTF file of the transcriptome assembly. The corresponding intronic sequences were retrieved from the human reference genome. The average number of introns per transcript was determined for each proto-gene, as well as their size, and were compared according to the age and the genomic position of the proto-gene, respectively.

From proto-genes containing at least one intron, we retrieved those from which the ORFs were located on one single exon and therefore not affected by splicing, and compared them according to their age and their genomic position.

Since we investigated if introns from “intergenic” proto-genes may result from the recycling of already existing human introns, we extracted all introns present in proto-genes from the “intergenic” genomic location, and randomly selected a subset of 500. The coordinates of the human introns from established genes were determined by using the The University of California Santa Cruz (UCSC) Table Browser [57], and the FASTA sequences of all annotated introns were retrieved accordingly.

The 500 randomly selected introns from “intergenic” proto-genes were used as query for a BLASTn search against all annotated human introns. We extracted all BLAST hits with an E-value ≤ 0.01, a percentage of ID > to 80%, and ≥80% length overlap between query and database sequence.

### 2.3. Motif Search

We investigated the presence of transcription factor motifs upstream of the transcription starting point of proto-genes. Transcription can be regulated at different levels: by promotors and by enhancer sequences [21]. We searched for transcription factor motifs by using a database from JASPAR [58], which contains 746 annotated vertebrate specific transcription factor binding motifs from promotors and enhancers. For simplicity, we refer to this database as “MotifsAllUpstream”.

We then scanned DNA regions locating 200 base pairs (bp) upstream of the transcription starting site (TSS) of proto-genes for the previously detected motifs; 200 bp upstream of the TSS covers the promotor and part of the proximal regulation region. The transcription starting points were accessed through the gtf file of the transcriptome assembly. The 200 bp sequences upstream of the TSS were downloaded for each of the 23,135 proto-genes without an unusual start codon (subset 1). Additionally, we retrieved the same regions for all established human coding genes (Ensembl = 21,445, subset 2), pseudogenes (Ensembl = 15,200, subset 3), and long non-coding RNA (lncRNA, Ensembl = 23,934, subset 4). The transcription starting points of the genes were retrieved from the GTF file of the human genome.

As a control for comparison, we searched these motifs in two more datasets: one dataset with non-coding regions and another dataset with intronic sequence. To build these datasets, we first calculated the positions of all non-coding regions and intronic regions in the human genome by using the GTF file and retrieving the corresponding FASTA sequences. Secondly, we randomly picked 23,135 regions (as many as the number of proto-genes) of 200 bp from all non-coding regions (subset 5) and intronic regions (subset 6). Overlaps between the selected regions were excluded, as well as overlaps with “intronic” and “intergenic” proto-genes. We compared the number of motifs found in the six subsets of upstream sequences.

As a second analysis, we also searched for motifs, specifically for core promotors, by using another JASPAR database containing nine motifs from core promotors. We refer to this database as “MotifsCore”. Such motifs are located between about 50 nucleotides before and after the TSS [59]. We therefore analysed DNA sequences from 100 bp before to 100 bp after the TSS of each proto-gene. We additionally looked at core motif regions (100 bp up and downstream of TSS) from established human coding genes, pseudogenes, and lncRNAs. We also searched for these regions in the previously randomly selected intronic and intergenic 200 bp sequences.

We searched for the presence of motifs from the two JASPAR motif databases in the respective six subsets of DNA sequences using the biopython package [60]. For each detected motif, a position-weight was calculated, followed by a position-specific scoring matrix (PSSM). We used PSSM score to search for the motif in our six datasets and selected a score threshold of 0.7 to obtain a large score window. We further filtered the detected motifs for 95% similarity with the reference matrix of the motif (Appendix A). In a second run, we decreased the similarity threshold to 80%, to investigate if a lower similarity might be a property of proto-gene motifs.

### 2.4. UTR Analyses

Each *de novo* transcribed ORF was mapped to its corresponding spliced transcript. The direction of the transcription (forward or reverse) was assessed, and accordingly the 5′ and 3′ UTRs were retrieved and stored in FASTA format. In order to compare the UTRs of proto-genes with those of established human genes, the coordinates of the UTRs of established human genes were extracted from the respective GTF file, and their corresponding FASTA sequences were downloaded from the human reference genome (GRCh38).

The size and GC content of UTRs were calculated for established human genes as well as for proto-genes. The structural and 2-dimensional properties of the 5′ UTRs were analysed with ViennaRNA 2.0 software [61]. The lowest minimum free energy (MFE) structure was calculated for all 5′ UTRs with the RNA fold function, which implements the algorithm of [62], yielding a single optimal structure.

We further predicted the equilibrium pair probabilities using the command “RNAfold-p-MEA”, which calculates base pair probabilities in the thermodynamic ensemble. This function allows us to access the frequency of the MFE structure, that corresponds to the probability of a structure to occur according to the Boltzmann weighted ensemble of all structures. The values of ensemble diversity were detected with a python script and correspond to the average Base pair distance between all structures in the thermodynamic ensemble. We calculated the structure probability and the ensemble diversity of 5′ UTRs of all proto-genes and compared it to the probability and diversity of 5′ UTRs of all human genes and lncRNA entire sequence. The lncRNAs do not contain UTRs, so we calculated instead the secondary structure of the entire sequences of the lncRNA sequences, by selecting those whose sizes ranged within the same sizes of the 5′ UTR of *de novo* and established human genes.

### 2.5. Domain Annotation

To compare the proteins encoded by proto-genes with proteins encoded by established human genes, we downloaded the 20,401 human genes from the database Ensembl and their corresponding transcripts. Annotated isoforms were analysed and the longest isoforms were kept in order to extract protein sequence.

The Pfam database and the pfam_script ‘scan.pl’ with default parameters [63] were used to annotate domains in both proto-proteins and established human proteins. For all proto-proteins, the results were sorted according to the age class and the genomic position of the proto-genes coding for the respective proteins.

A Gene Ontology (GO) term enrichment was carried out with *topGO* package in R [64]. Domains were annotated with the ‘pfam2go’ mapping of *Pfam* domains to GO terms [65]. The enrichment analysis was carried out using the ontologies of ‘Molecular function’ and ‘Biological process’ with *topGO*s ‘weight01’ algorithms. Significantly enriched (*p*-value ≤ 0.05) GO terms are visualized as tag clouds.

For detection of sequences with the potential to code for novel domains in the studied proto-proteins, the software SEG-HCA was used [66]. The software divides protein sequences by gathering the regions with strong hydrophobic residues and linkers. The highly hydrophobic clusters are known to give rise to secondary structures, which may evolve into new protein domains. The oldest proto-proteins (I5) were not analysed regarding their hydrophobic clusters, as they contain annotated domains which would bias the analysis.

### 2.6. Statistical and Bioinformatic Analyses

All data extractions and analyses were performed with python. The developed scripts are freely available at https://github.com/AnnaGrBio/De_novoGenes_NewGenomicSignals (accessed on 16 December 2021). Statistical analyses and graphs were performed in R (R Core Team; 2017). Since all datasets contained between 500 and 30,000 values, Shapiro tests were not necessary to test normality. According to the equality of variances (Fisher test), we used Welch student test and standard two sample student test for parametric two sample assessment. Anova or Kruskal-Wallis tests were used for multiple sample test comparisons. Linear regressions were used for consideration of the age classes as time variations.

## 3. Results

A total of 27,550 human proto-genes were retrieved from the dataset created by [33]. We only considered proto-genes with a minimum expression strength of Reads Per Million Mapped reads (RPM) > 0.5. We found 4415 ORFs from proto-genes which began or ended with an unusual codon or which were not assigned to the correct genomic position. The corresponding proto-genes were excluded from the analysis (except for the domain analyses), giving rise to a total number of 23,135 human proto-genes.

### 3.1. Introns

#### 3.1.1. Introns in Proto-Genes

The number of proto-genes containing introns was calculated, and these numbers were compared according to the genomic position and age class of the proto-gene (Figure 2). Contrary to our expectations, the number of proto-genes containing introns was surprisingly high, as introns were found in 15,710 (67.9%) of the 23,135 proto-genes. According to their age class, 75–100% of “exonic” proto-genes possess at least one intron (Figure 2a). The proportions are lower for the other two genomic positions, and vary between 10–30% for “intronic” proto-genes, and between 2.6–30% for “intergenic” proto-genes. For each genomic position, we observed an increase in the proportion of proto-genes containing introns compared to the older age class of proto-gene. The older a proto-gene, the more the probability of it containing at least one intron (Table 1).

Putative proto-genes contain significantly less introns than established human genes (established genes: 6.57; proto-genes: 5.72; student test *p*-value < 2.2 × 10^−16^). For all age classes, the average number of introns is higher in “exonic” proto-genes compared to “intronic” and “intergenic” proto-genes (Figure 2b). Comparing the average number of introns in “intronic” and “intergenic” proto-genes, we found fewer introns in “intergenic” proto-genes for age classes I0 to I4 (Table 2).

For “intergenic” and “exonic” proto-genes, the average number of introns increases significantly with gene age. This increase is much stronger in the “exonic” proto-genes than in the “intergenic” genes (“exonic” slope lm = 0.50427, lm = 0.20548 without I5, “intergenic” slope lm = 0.25627), but remains low and stable in proto-genes found in the “intronic” genomic position (slope lm = 0.02916).

Among the 15710 proto-genes with introns and at least two exons, 2148 were found to have the regarding ORF located on one single exon (Figure 2c). The proportion of ORFs found in one single exon is higher for proto-genes with the genomic positions “intergenic” (60.16%) and “intronic” (63.36%) than for “exonic” (36.14%). Moreover, taking the gene age into consideration, we found a lower proportion of ORFs located on one single exon compared to ORFs spread over multiple exons with increasing age in all genomic positions (“exonic”: from 37.78% (I0) to 3.93% (I5); “intergenic”: from 76.19% (I0) to 18.12 (I5); “intronic”: from 73.45% (I0) to 49.24 (I5); Figure 2d).

#### 3.1.2. Origin of Introns in “Intergenic” Proto-Genes

To investigate the origin of introns in “intergenic” proto-genes, 500 randomly chosen introns from “intergenic” proto-genes were compared via BLASTn 2.7.1 with a dataset of established human introns. 205 of the 500 *de novo* introns were found to have a reasonable hit with at least one established human intron (41%). All other tested *de novo* introns (59%) did not match with annotated introns, suggesting that they emerged by intronisation. Among the 205 *de novo* introns also found in established human genes, 37 were found in proto-genes whose ORF was located on one single exon.

### 3.2. Motifs Upstream Proto-Genes

#### 3.2.1. Average Number of Motifs from “MotifsAllUpstream”

We searched for 746 motifs from the databases “MotifsAllUpstream” (see method) in six datasets of sequences (from different genomic locations: 200 bp upstream of proto-genes, 200 bp upstream established genes, 200 bp upstream of pseudogenes, 200 bp upstream of lncRNA, 200 bp in randomly selected intergenic regions, 200 bp in randomly selected introns of established genes). For the motifs found upstream of proto-genes, the results were analysed according to the different age classes and genomic positions of proto-genes. A motif is considered as present if it shows 80% identity with its reference position weight matrix (see Methods). Results for a more a common threshold of 95% are available in the Appendix A, and significantly show the same trends.

Surprisingly, the average number of motifs (“MotifsAllUpstream”) per sequence is lower in the upstream region of established genes compared to the upstream region of proto-genes (Welch *t* test, *p*-value = 2.2 × 10^−16^), with the latter showing a lower average number of motifs compared to the four other datasets (pseudogenes, lncRNA, intergenic regions, introns) (Figure 3a) (Welch *t* test, *p*-value = 2.2 × 10^−16^ for the four comparisons). We investigated the distribution of the 60 most common motifs over the six datasets (Figure 3b). For 56 of the 60 motifs we observed either the lowest number of motifs for established genes, followed by the dataset of proto-genes, and the highest number for the four other categories, or we observed the exact opposite: the highest number of motifs for established genes, followed by the dataset of proto-genes, and lastly the four other categories with lower number of motifs.

The database “MotifsAllUpstream” contains both motifs from promotors (except core promotor) and enhancers. In order to distribute the motifs in these two categories, we studied their distribution in 200 bp upstream of all human genes. The motifs most present 200 bp upstream of established genes were considered specific for promotors. The remaining ones are considered specific for enhancers (or distal promotors). We accordingly partitioned the database “MotifsAllUpstream” into two databases, that we call “AllPromotor” and “Enhancer”. We calculated the average number of motifs from “AllPromotor” and “Enhancer” in the region 200 bp upstream of proto-genes, according to their age class and genomic position.

Interestingly, proto-genes from the “exonic” genomic position have more motifs specific to the “AllPromotor” dataset, than proto-genes from “intronic” and “intergenic” positions (Welch *t* test, *p*-value = 2.2 × 10^−16^ for the two tests) (Figure 3c). On the contrary, the “intergenic” and “intronic” proto-genes have more motifs specific to “Enhancer” compared to “exonic” proto-genes (Welch *t* test, *p*-value = 2.2 × 10^−16^ for the two tests). These results were not affected by the age class of proto-genes, except for motifs from the database “AllPromotor” found in “exonic” proto-genes. In this specific case, the average number of motifs is higher for “I5” proto-genes than for “I0” proto-genes (Welch *t* test, *p*-value = 0.000656).

As “intergenic” and “intronic” proto-genes are enriched in enhancer motifs compared to “exonic” proto-genes, we also wondered if this enrichment was higher than expected by chance. We therefore compared the average number of motifs from the “AllEnhancer” database found in “intronic” and “intergenic” proto-genes, to the average number of the same motifs found in the 20,000 non-coding regions randomly selected. We discovered that “intergenic” and “intronic” proto-genes are significantly enriched in enhancer motifs compared to non-coding regions, with “intergenic” proto-genes being most enriched (mean motifs “intergenic” proto-genes: 330.7; mean motifs “intronic” proto-genes: 319.8, mean motifs non-coding regions: 305.8; *t* test “intergenic” proto-genes vs. non-coding region: *p*-value = 2.2 × 10^−16^; *t* test “intronic” proto-genes vs. non coding-region: *p*-value = 1.579 × 10^−8^). We investigated the function of the 50 most frequent enhancer motifs found upstream “intergenic” proto-genes. The large majority of these motifs (48/50) were found to be associated to homeo genes, which are well known to be involved in development (Appendix A).

#### 3.2.2. Distribution of “MotifsCore” Motifs

We searched for the nine motifs of the database “MotifsCore” (specific from core promotor), in six datasets of sequences (200 bp surrounding transcription starting point of proto-genes, established genes, pseudogenes, lncRNA, and 200 bp in randomly selected intergenic regions and introns of established genes).

The average number of transcription factor motifs from the “MotifsCore” database is highest in promotor regions of established genes (mean = 34.64) compared to promotor regions of proto-genes (mean = 32.20; *t* test *p*-value = 2.2 × 10^−16^), and highest in promotor regions of proto-genes than in the four other datasets (mean lncRNA 29.97; mean pseudogenes 30.06; mean intron 29.79; mean non-coding region 29.97; *t* test, *p*-value = 2.2 × 10^−16^ for the 4 tests; Figure 3d).

Next, we investigated whether the average number of motifs from the “MotifsCore” database was divergent according to the age class and the genomic position of proto-genes (Figure 4). We observed that proto-genes at the “exonic” genomic position showed an enriched number of motifs compared to “intronic” and “intergenic” proto-genes (Welch *t* test with average motif numbers, *p*-value < 2.2 × 10^−16^). Meanwhile, no significant differences in the numbers of motifs of “intronic” and “intergenic” proto-genes were found (Welch *t* test for average motif numbers, *p*-value = 0.2962), for all age classes. By comparing motif numbers of “exonic” proto-genes with motif numbers of established genes, we found that “exonic” proto-genes contain fewer core motifs in age class I0 than established genes (32.8 vs. 33.6), but this difference is not significant. In age class I5, the average number of motifs in “exonic” proto-genes is similar to the average number of motifs found in established genes (33.5 vs. 33.6).

The observed data did not follow a normal distribution (Shapiro tests, *p*-value 2.2 × 10^−16^). We observe two interesting phenomena when investigating the motif distribution in regions upstream of proto-genes. First, in the upstream region of “exonic” proto-genes, we found that the density curve showed two different peaks for proto-genes I0, I1, I2, I3 and I4. These results suggest that the “exonic” proto-genes in these age classes may fall into two categories, one being more enriched in motifs, and the other comprising significantly fewer. Secondly, in “intergenic” proto-genes from age classes I2, I3, I4 and I5, we observed a bulge in the curve of distribution, suggesting the same conclusion.

Finally, we calculated the percentage of proto-genes which had each of the nine motifs present in their promotor, for each age and genomic position. As expected from previous results, “exonic” proto-genes have on average a higher percentage of all the motifs compared to “intronic” and “intergenic” proto-genes (Figure 5). However, we found the opposite trend with the TATA box motif.

### 3.3. UTRs of Proto-Genes

#### 3.3.1. Size and GC Content of UTRs

5′ UTRs and 3′ UTRs of human proto-genes were compared to those of established human genes. We observed that the size of the 5′ UTRs tends to be longer in the youngest proto-genes (I0) compared to established human genes, and become shorter with age (Appendix A).

We found an average GC content of 59.26% in the 5′ UTR of the human established genes, which is consistent with the literature [67]. The average GC content in the 5′ UTRs of proto-genes was 53.23% on average, and is significantly lower than in established genes (Welch *t* test *p*-value 2.2 × 10^−16^), (Figure 6a).

For 3′ UTRs we find the opposite trend: the GC content of 3′ UTRs of established genes is significantly lower (mean: 45.20442), compared to 3′ UTRs of proto-genes (mean: 47.1894, *t* test *p*-value 2.2 × 10^−16^; Figure 6b).

We observed that the GC content was significantly higher in the 5′ UTRs of “exonic” proto-genes compared to “intronic” and “intergenic” proto-genes for all age classes (*t* test, *p*-value = 2.2 × 10^−16^ for the two tests) (Figure 6c). Moreover, the average GC content did not vary between age classes for “intronic” and “intergenic” proto-genes, whereas the GC content of “exonic” proto-genes is significantly higher in the oldest proto-genes than in younger proto-genes (Welch *t* test, *p*-value = 0.005916).

#### 3.3.2. 2D Structure of the 5′ UTRs

We studied the secondary structure of the 5′ UTRs of human proto-genes and established human genes, and used lncRNA sequences as a control, as lncRNAs are generally not binding ribosomes. We compared the energy of the structures between proto-genes and established genes, to see if the UTRs of proto-genes are less or equally stable compared to those of established genes. Moreover, we compared the diversity and the probability of secondary structures of proto-genes, established genes, and lncRNAs.

The absolute value of the minimum free energy of the thermodynamic ensemble of all 5′ UTRs decreases with the size of the 5′ UTRs, indicating a higher stability of the structure (Figure 7a, left, Appendix A). Most of the 5′ UTR sizes ranged between 0 and 200 bp. The longer a sequence, the lower folding free energy it needs, which provides better stability. At equivalent sizes, however, established genes’ 5′ UTRs use lower folding free energy than proto-genes, suggesting a higher instability in proto-gene structure. (Figure 7a; lm model, established genes: intercept: 6.77884, slope: −0.40600; proto-genes: intercept: 5.04931, slope: −0.33856).

We observed that the structure prediction frequencies (structure probability) are significantly higher for 5′ UTRs of established genes compared to 5′ UTRs of proto-genes (*t* test, *p*-value < 2.2 × 10^−16^), but interestingly the structure prediction frequency is higher in proto-genes compared to lncRNA (*t* test, *p*-value < 2.2 × 10^−16^) (Figure 7b). These results indicate that 5′ UTRs of proto-genes are less likely to fold into stable secondary structures compared to established 5′ UTRs, but more likely to fold than lncRNA sequences. Unexpectedly, the structure prediction frequency is higher in the 5′ UTR of “intronic” proto-genes (mean = 0.15) and “intergenic” proto-genes (mean = 0.15) than in “exonic” proto-genes (mean = 0.10) (*t* test “exonic” vs. “intergenic”, *p*-value = 2.2 × 10^−16^; *t* test “exonic” vs. “intronic”, *p*-value = 2.2 × 10^−16^; *t* test “intronic” vs. “intergenic”, *p*-value = 0.6133). Additionally, we found that the average diversity value, which indicates the instability of a predicted structure, was significantly higher in proto-genes than in established genes (*t* test, *p*-value < 2.2 × 10^−16^), but was significantly lower in 5′ UTRs of proto-genes than in lncRNA (*t* test, *p*-value < 2.2 × 10^−16^; Figure 7c). Therefore, 5′ UTRs of proto-genes are less stable than 5′ UTRs of established genes, but more stable than lncRNA. We also observed that the diversity of the structure was higher in “exonic” proto-genes (mean 62.26) than in “intronic” proto-genes (mean = 54.85) and “intergenic” proto-genes (mean = 55.84) (*t* test “exonic” vs. “intergenic”, *p*-value = 3.181 × 10^−9^; *t* test “exonic” vs. “intronic”, *p*-value = 5.074 × 10^−12^; *t* test “intronic” vs. “intergenic”, *p*-value = 0.4401). The tendencies for structure probability and diversity (stability) therefore complement each other for all analysed gene/RNA classes and genomic locations: the more likely it is for a UTR to have a structure, the more stable it seems to be.

### 3.4. Domains in Proto-Genes

#### 3.4.1. Annotated Domains

We compared the absolute number of Pfam domains found in established human proteins and human proto-proteins; 11,244 established human genes contained domains (55%), while only 7223 proto-proteins contained at least one annotated domain (26%).

We then investigated if the age and the genomic position of proto-genes correlate with the presence of annotated domains in the corresponding proto-protein. Independent of the genomic position (“exonic”, “intronic”, “intergenic”), we did not find any known domains in proto-proteins at ages I0, I1 and I2. Only 0.07% of proto-proteins contain an annotated domain in age class I3, and 0.22% in age class I4 (Appendix A). More than 99.7% of the domains found in proto-proteins are in fact present in the oldest age class I5, which are shared with mouse (Figure 8a).

Half of the proto-proteins from the age class I5 (49.02%) contain an annotated domain, of which 96.30% are located at the “exonic” genomic position (Appendix A). We still found 185 proto-proteins from I5 and the “intronic” genomic position, and 82 proto-proteins from I5 appeared in the”intergenic” genomic position, containing at least one annotated domain. Taken together, these results indicate that more than 98% of all proto-proteins that possess a known domain are the oldest proto-proteins (I5; shared with mouse ) and are overlapping with an exon. 99.96% of proto-proteins which are younger than I5 and come from the “intronic” or “intergenic” genomic position do not contain any known domain.

#### 3.4.2. Novel Domains in Proto-Proteins

Except for the oldest proto-genes (I5) overlapping with an exon (“exonic”), most proto-proteins did not contain any known domain. We hypothesise that proto-proteins become functional over time by gaining novel domains. We used Seg_HCA [66] to search for hydrophobic clusters as an indicator for existing or emerging domains in proto-proteins from the age classes I0 to I4. The oldest age class I5 was excluded from the analysis, in order to avoid a bias due to the presence of known domains. For each proto-protein, we retrieved the number of hydrophobic clusters found in the sequence, as well as their size. We found hydrophobic clusters in proto-proteins of all studied age classes (I0–I4, Figure 8b). The number of hydrophobic clusters is significantly lower in the age class I0 than in all other age classes (Welch *t* tests (I0–I1; I0–I2; I0–I3; I0–I4, *p*-values between 2.2 × 10^−^^16^ and 8.238 × 10^−^^9^). The number of hydrophobic clusters is also significantly lower in I2 and I4 compared to I5 (Welch *t* tests, *p*-value = 0.0022; *p*-value = 0.00046). The average number of hydrophobic clusters increases with age (linear model, slope = 0.1732102, mean I0 = 1.300525; mean I1 = 1.473735; mean I2 = 1.5208; mean I3 = 1.47424; mean I4 = 1.573866).

The average number of hydrophobic clusters is higher in proto-proteins from the genomic position “exonic” (mean = 2.52) than from the genomic positions “intronic” (mean = 1.44; welch *t* test *p*-value = 2.2 × 10^−16^) and “intergenic” (mean = 1.34; welch *t* test *p*-value = 2.2 × 10^−16^) for all analysed age classes (Figure 8c). This was observed for any age class (statistical tests are shown in Appendix A). “Intronic” proto-proteins have more hydrophobic clusters than “intergenic” proto-proteins (Welch *t* test, *p*-value = 1.526 × 10^−^^10^), but these differences are only significant for age classes I2 and I4. The difference between number of hydrophobic clusters in “intergenic” and “intronic” proto-genes was lower compared to “exonic” proto-proteins (statistical tests are shown in Appendix A).

## 4. Discussion

In the present manuscript, we investigated four different gene properties of 23,135 human proto-genes: presence of introns, presence of motifs, characteristics of UTRs and domain coverage. All proto-genes were investigated according to their genomic position and their age.

### 4.1. Introns

In 2011, Wu et al. (2011) [39] reported that 59 of the 60 *de novo* genes identified in humans are single exon genes. Another experimental study investigated the function of *de novo* genes and reported that most of the analyzed *de novo* genes contain a single exon [5]. In contrast, we found that half of all human proto-genes of our dataset contained introns. This very unexpected result suggests a more complex structure than previously thought for these new genes, even though proto-genes contain fewer introns than established genes. We found that “exonic” putative proto-genes contain on average more introns than “intergenic” and “intronic” proto-genes. A possible explanation could be that “exonic” proto-genes not only overlap with parts of an established gene but copied introns of the established gene into their own sequence. Our finding that about 50% of the analysed proto-genes contain introns indicates that splicing of proto-genes, surprisingly, already occurs even in the youngest proto-genes. This would explain why Dowling et al. (2020) [33] found multiple transcripts per ORF with differing lengths during their transcriptome assembly.

Furthermore, we discovered that 13.7% of the analysed human proto-genes containing introns have ORFs located on one single exon. Such a structure is uncommon but has already been documented in the fruit fly genome, in the *de novo* gene *atlas* [68]. The coding sequence of *atlas* is located on the first of the two exons present in the gene. Our study suggests that this structure is more common than expected in proto-genes. ORFs located on one single exon are more common in young than in old proto-genes. It has already been reported that *de novo* ORFs increase in length over time [33], which could explain why small ORFs located on one single exon evolve to be overlapping with several exons by gaining mutations.

We found that the average number of introns increases with age for all proto-genes. This result suggests that proto-genes acquire introns over time. Seven scenarios are known to describe intron emergence in genes [38]: intron transposition, transposon insertion, tandem duplication, intron gain during double strand break repair, insertion of a group II intron, intron transfer, and intronisation. Interestingly, 41% of the introns from a subset of 500 randomly picked introns from “intergenic” proto-genes had a BLAST hit with established introns from the human reference genome. The origin of introns in proto-genes has already been investigated in Oryza species. In 2019, Zhang et al. (2019) [69] studied high quality genomes of 13 Oryza species to study *de novo* origination of proto-genes. The authors conducted a study of 175 *de novo* genes which contained 362 introns. The authors further concluded that these introns were directly recruited from the non-coding ancestral intergenic DNA. In our study, 59% of investigated introns show no detectable signal of homology to existing introns using BLAST. This finding suggests that proto-genes captured introns from non-coding DNA, which is in agreement with the findings of [69]. However, 41% of the investigated introns have a detectable signal of homology to existing introns, suggesting that some proto-genes also captured existing introns. It would also be interesting to focus on the putative role that transposable elements may have played in proto-gene emergence. It has already been shown that orphan genes can emerge via the exaptation of transposable elements [6]. In 2021, Jin et al. (2021) [70] discovered that 37 orphan genes in the genome of *Oryza sativa* had emerged via recruitment of transposable elements. The presence of introns in “intergenic” proto-genes could also be an interesting indicator that proto-genes and/or their introns emerged via the movement of a transposable element.

### 4.2. Motifs

By comparing four datasets of sequences to upstream regions of established human genes and human proto-genes, we demonstrated that “exonic” proto-genes are significantly enriched in motifs which are characteristic for promotors and core promotors. The average number of motifs specific for promotors found upstream of “exonic” proto-genes still remains lower than the number of promotor motifs found upstream of established human genes, except for the oldest “exonic” proto-genes. No difference was found between the average number of core promotor motifs of established human genes and “exonic” proto-genes at the age class I5. As these old proto-genes overlap with established human genes, this finding suggests a reuse of pre-existing regulatory sequences in old proto-genes [71]. In 2015, [72] observed an enrichment in transcription factor binding sites in the promotor regions of the 2714 proto-genes that they identified in humans and chimpanzees: 20% of these proto-genes matched with long lncRNAs, and half of the dataset overlapped with exons and partly with introns, which may have biased the results. The observed increase of transcription factor binding sites in core promotor motifs with the age of “exonic” proto-genes may indicate a gain of motifs with age. The “exonic” proto-genes are the most complicated to characterize, as they may or may not overlap with the promotor and can also be transcribed in the reverse direction.

On the other hand, we demonstrated that “intronic” and “intergenic” proto-genes are specifically enriched in enhancer motifs, and that their average number of enhancer motifs is higher than in randomly selected non-coding regions. This means that their proximity to these motifs might be a condition for their pervasive existence. The enrichment of enhancer motifs in proto-genes compared to non-coding regions, however, does not change with age, corroborating their “sine qua non” status.

The fact that putative “intergenic” and “intronic” proto-genes contain more enhancers than promotor motifs is an interesting finding. One of the reasons could be that non-coding regions, such as intergenic and intronic regions, may be translated through existing enhancers rather than gaining their own regulatory motifs. This result confirms the finding of [73], who demonstrated that young *de novo* genes in mouse emerge closer to enhancers than to promotors. They assumed that *de novo* transcribed ORFs, which emerge in non-coding regions, may be transcribed due to the facilitated environment created by the DNA loop. They also found that older *de novo* genes were in closer proximity to promotors than to enhancers. In contrast to this, we did not find any significant change with gene age. However, [73] did not distinguish between “intergenic” and “exonic” *de novo* genes except for the youngest age class, which makes the studies not entirely comparable.

Not only do “intergenic” and “intronic” proto-genes appear more often in regions rich in enhancers, but most of these enhancers are associated with morphogenesis functions. Homeodomain proteins are highly expressed [74], and can regulate the expression of many genes [75]. The combination of these two properties could explain why any new ORF which emerges close to homeotic transcription factors has a higher chance of being expressed and maintained throughout evolution.

The average number of motifs specific for core promotors found upstream of “intergenic” and “intronic” proto-genes is low and does not significantly differ from non-coding-regions. However, when looking at the distribution, we observed subgroups enriched in motifs from the core promotor compared to the remainder of the group. Motifs from the core promotor are more frequent in a subset of “intergenic” proto-genes compared to the average trend. This result might confirm that “intergenic” proto-genes can gain promotor motifs with age. Again, further investigations using a wider range of data sets is warranted to confirm the generality of these findings.

Even though “intergenic” and “intronic” proto-genes are not enriched in core promotor motifs, the TATA motif was found to be more present upstream of these proto-genes than upstream of “exonic” genes. Generally, not all established human genes have TATA motifs in their promotor. According to [76], only 24% of human genes do so. Genes containing a TATA box in their promotor have been shown to show increased transcriptional regulation compared to genes without a TATA box [77]. In 2013, [78] showed that genes from *S. cerevisiae* with an essential TATA box have a significant codon bias and higher levels of expression. We hypothesise that the presence of a TATA box in “intergenic” or “intronic” proto-genes could be maintained by selection, as this allows the expression of the gene. The enrichment in TATA boxes and homeobox transcription factors upstream of “intergenic” and “intronic” proto-genes could also be an explanation for their pervasive expression.

### 4.3. UTRs

We investigated the size, GC content, and the predicted free energy of the 2D structure of 5′ UTRs in prot-genic mRNA and compared these properties to those to established genes. 5′ and 3′ UTRs are longer in proto-genes than in established genes, except for “I5”. Until now, studies have focused only on increasing length of *de novo* ORFs with age, which can be due to the gain of a new start or stop codon in the transcript. In cases in which the length of an ORF is not modified during evolution, the length by which a transcript increases over time correlates with a decrease in length of its 5′ UTR. In 2009, [79] showed that mammalian 5′ UTRs evolve under weaker selective constraints, giving rise to an accumulation of new AUG codons. In *S. cerevisiae*, [80] found that lineages that had a longer 5′ UTR showed a substantial decrease of expression correlation with ribosomes. Moreover, Reuter et al. (2008) [81] observed a correlation between the size of 5′ UTR regions and the GC content. The authors concluded that the size of UTRs may therefore be shaped by structural forces.

We observed that the GC content of the 5′ UTRs is significantly lower in proto-genes than in established genes. However, the GC content is higher in “exonic” proto-genes than in “intergenic” and “intronic” proto-genes. A large number of “exonic” proto-genes have a 5′ UTR overlapping with established exons, and generally have higher GC contents compared to non-coding regions [82]. Hence, the emergence of proto-genes in regions of high GC content may promote higher levels of structure in the 5′ UTR. In contrast, the emergence of new genes overlapping with introns or non-coding regions results in 5′ UTRs that have a low GC content.

In a second step, we compared the structure diversity and probability of the UTRs of proto-genes with the same properties of established genes and lncRNA. As expected, the probability of forming stable RNA structure was lower in UTRs of proto-genes than in established genes, and the diversity of the ensembl structure was higher in proto-genes. However, we observed the opposite trend in lncRNA, indicating a higher probability in UTRs of proto-genes compared to lncRNAs. This result suggests that the foldability and the structure of 5′ UTRs of proto-genes could be prone to selection.

Taken together, proto-genes show a lower GC content, a lower probability to form stable RNA structure from their UTRs, and a higher number of discorded regions (structure diversity) compared to established genes, but a higher structure probability and lower likelihood for disordered regions than lncRNAs. We therefore hypothesise that 5′ UTRs of proto-genes are less likely to fold compared to 5′ UTRs of established genes but are more likely to fold compared to lncRNAs. Surprisingly, we found that UTRs of “intergenic” and “intronic” proto-genes are more likely to fold stably than UTRs of “exonic” proto-genes. Hence, we found the UTRs of ”intergenic” and “intronic” proto-genes have a low GC content but a higher potential for folding. Low GC content is a characteristic of protein binding mRNAs [83]. RNA interference methods have also shown that low GC content can increase RNAi efficiency [84]. It is possible that the UTRs of “exonic” proto-genes are constrained to have a high GC content as they overlap with coding exons, whereas intergenic and intronic UTRs are subjected by fewer constraints.

### 4.4. Domains

26.21% of the analysed proto-proteins contained annotated domains. Among these, 96.30% were found in “exonic” proto-genes, and nearly all of them belong exclusively to the oldest outgroup proto-genes (I5). In other words, the majority of all detected known/annotated domains within proto-genes originated at least 90 Mya and share a common ancestor with mouse. Young proto-genes do not contain known/annotated domains in their encoded proteins. Furthermore, primate specific “intronic” and “intergenic” proto-proteins do not contain any detectable pfam domains (I0 to I4). We highlight that the absence of annotated domains can be used as a support to distinguish any proto-genes from any other novel gene (emerging by other mechanism than “*de novo*”).

However, this criterion does not work for old proto-genes. One reason why only I5 “exonic” proto-genes contain known domains could be that “exonic” proto-genes are able to gain domains from the exon with which they overlap. Mechanisms have been described to illustrate how proteins can gain domains [49,85]. Notably, exon extension allows proteins to gain pre-existing domains [54,86]. This fits in with the observation that *de novo* genes become longer with age [33]. The increase in length and the mutations throughout evolution could allow new genes to capture existing domains from their host gene. More interesting is why young (I0 to I4) “exonic” proto-genes never contain domains, although they could utilise a present domain of the overlapping exon. This could be due to destabilising effects of the added sequence parts to the existing domain, rendering it non-functional [87]. In 2013, a study [54] used Pfam domains to investigate their emergence time, dividing them into “mammal specific”, “vertebrate specific”, and “older domains”. In another study [55], novel domains in insects with divergence times between 2 My and 390 My were analysed. The authors demonstrated that novel domains could emerge from scratch in terminal regions of existing proteins.

As a next step, we investigated hydrophobic clusters to identify putative novel domains in protein from proto-genes aged I0 to I4 using SEG-HCA. We observed that the majority of proto-proteins contain hydrophobic clusters. Proteins from proto-genes emerging at the “exonic” genomic position contain more clusters on average than “intronic” and “intergenic” proto-proteins. Moreover, we show that the average number of hydrophobic clusters in proteins increases with age, and this increase is even stronger for “exonic” proto-genes. The observation that proto-genes contain hydrophobic clusters is in agreement with a recent study, in which the authors demonstrated that novel domains were present and evolved in yeast *de novo* genes [88]. Even if domains are conserved as independent units through evolutionary time and are mainly rearranged, novel domains have been shown to emerge regularly in proteins in order to add specific novel functionality [52,55].

## 5. Conclusions

The four studied properties (introns, transcription motifs, UTRs, and protein domains) show significant differences between proto-genes and established genes, making them exciting features when assessing the emergence of proto-genes. However, we demonstrate that these properties are not only strongly influenced by gene age, but also by genomic position of the proto-gene. The discovery that genomic positions affect proto-genes’ properties is of major importance. Future studies of proto-genes should therefore take the genomic position of proto-genes as well as their age into account.

## Figures and Tables

**Figure 1 genes-13-00284-f001:**
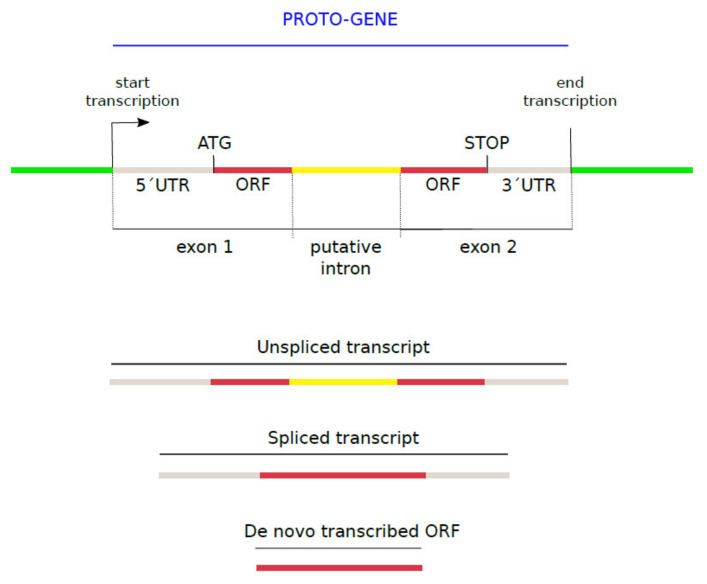
Structure of a proto-gene.

**Figure 2 genes-13-00284-f002:**
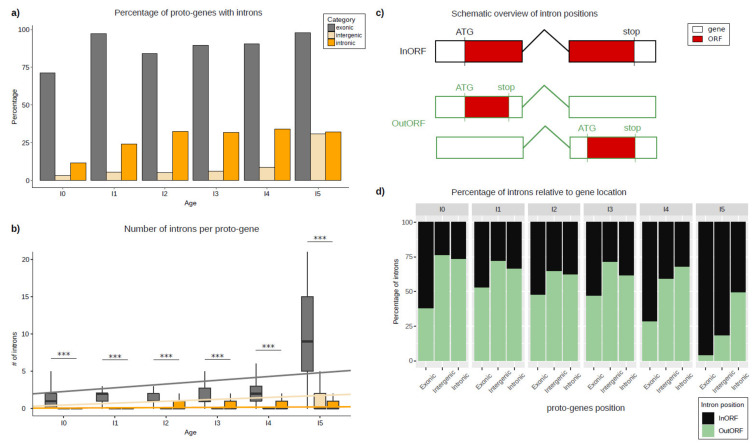
Introns in proto-genes. (**a**) Percentage of proto-genes containing at least one intron, shown for each age class and genomic position. (**b**) Number of introns per proto-gene, shown for each age class and genomic position. Regression models are shown for each genomic position. (**c**) Schematic overview of intron positions. Black/green boxes indicate separate exons, building a gene together with one intron, indicated by black/green, angled line. Red staining within exon(s) shows the according ORF. (**d**) Percentages of ORFs containing an intron (“InORF”, black) and ORFs not affected by splicing because they are located on one single exon (“OutORF”, green). Asterisks indicate significant differences between the groups (*** *p* < 0.001).

**Figure 3 genes-13-00284-f003:**
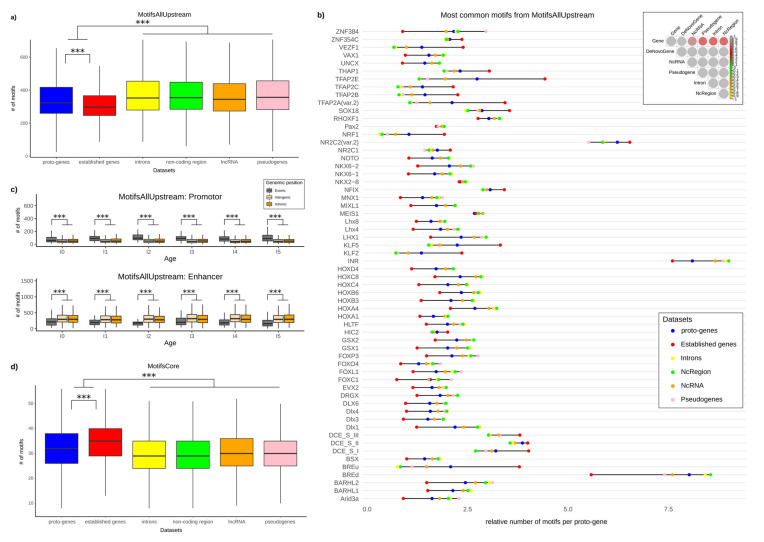
Number of motifs in proto-genes, (**a**) Number of motifs upstream of TSS (“MotifsAllUpstream”). Blue: motifs found in sequences 200 bp upstream of human proto-genes; red: motifs found in sequences 200 bp upstream of human established genes; yellow: motifs found in randomly selected human introns which do not contain proto-genes (200 bp length); green: motifs found in randomly selected non-coding regions which do not contain proto-genes (200 bp length); orange: motifs found in sequences 200 bp upstream of human lncRNA; pink: motifs found in sequences 200 bp upstream of human pseudogenes. The y-axis represents the average number of motifs per sequence found in the six datasets, with a similarity threshold of 80% to the consensus motifs. (**b**) Most common motifs from the database “MotifsAllUpstream”. The colours of the dots indicate the subset the found motifs were detected in. The y-axis represents the 60 most common motifs in the six datasets (see Figure 3a). (**c**) Average number of motifs from “MotifsAllUpstream” database found in proto-genes, shown for each age class and genomic position. The motifs are classified into two databases: “Promotor” (top) and “Enhancer” (bottom). The y-axes represent the average number of motifs. (**d**) Number of core motifs (“MotifsCore”). The colours indicate the different subsets of genetic sequences (see Figure 3a). The y-axis represents the average number of motifs found in the six datasets, with a similarity threshold of 80% to the consensus motifs. Asterisks indicate significant differences between the groups (*** *p* < 0.001).

**Figure 4 genes-13-00284-f004:**
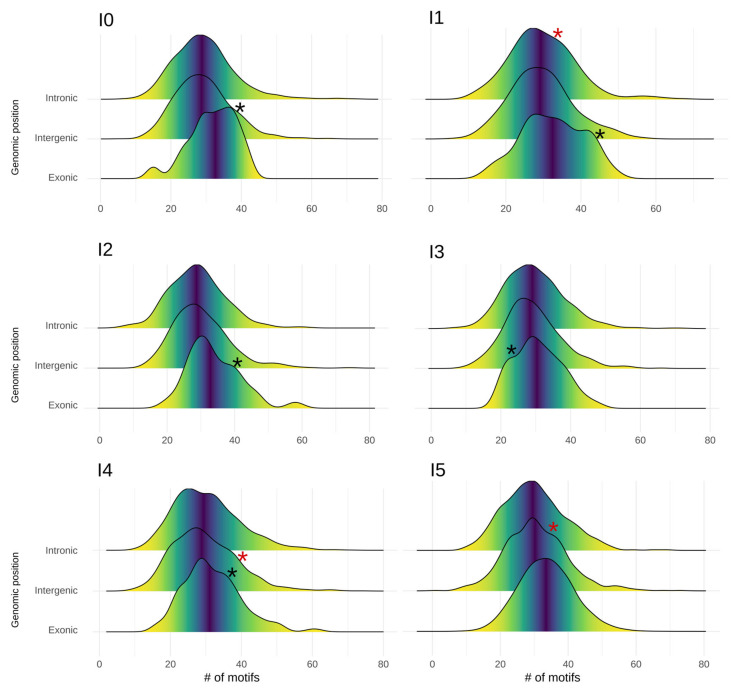
Distribution of core promotor motifs in proto-genes, shown for each age class (plots I0 to I5) and genomic position (lines). Each plot shows the frequency distribution of core motifs for “exonic” (low line), “intergenic” (middle line), and “intronic” (top line) proto-genes. Black asterisks represent an extra peak compared to a normal distribution. Red asterisks represent a bulge compared to normal distribution.

**Figure 5 genes-13-00284-f005:**
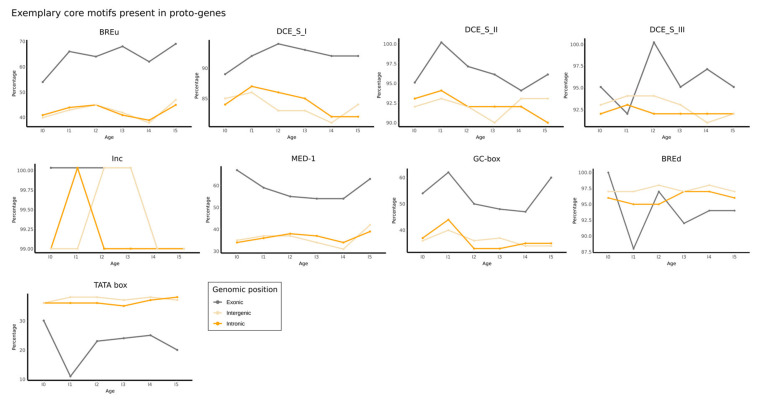
Exemplary core motifs present in proto-genes. Each plot represents one of the nine core motifs. Percentages of motifs present in putative human/de novo/genes are shown for each age and genomic position.

**Figure 6 genes-13-00284-f006:**
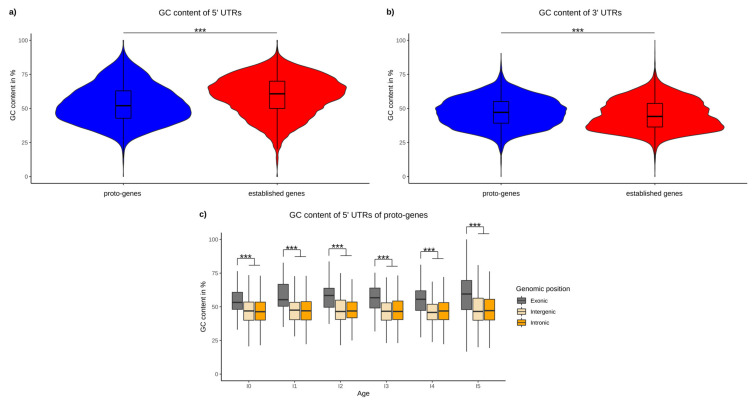
GC content of UTRs. (**a**) Average GC content of 5′ UTRs of proto-genes (blue) and human established genes (red). (**b**) Average GC content of 3′ UTRs of proto-genes (blue) and human established genes (red). (**c**) GC content of proto-genes according to their age and genomic positions. Asterisks indicate significant differences between the groups (*** *p* < 0.001).

**Figure 7 genes-13-00284-f007:**
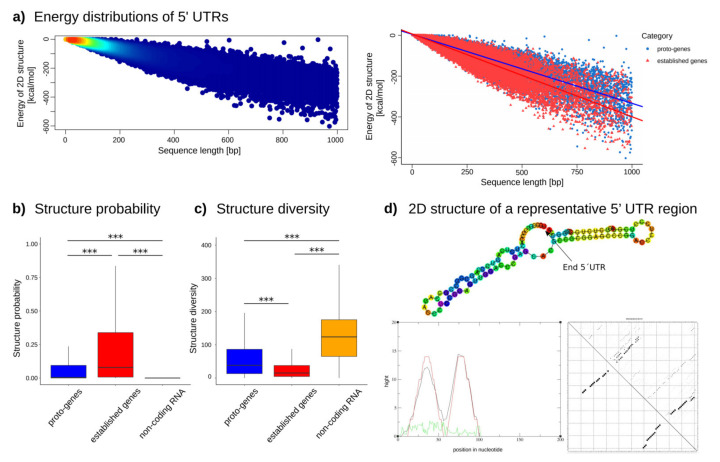
Properties of the 2D structure of 5′ UTRs in established genes and proto-genes. (**a**) Energy distribution of 2D structures. The left plot shows the energy distribution of 2D structures of proto-genes and human established genes, indicating the frequency of genes with the same size and energy, with a high density in red and a low density in blue. The right plot shows the energy distribution of 2D structures of proto-genes in blue and of established human genes in red. (**b**) Structure probability of human proto-genes (blue), established human genes (red) and human lncRNAs (orange). The y-axis represents the probability of the structure. (**c**) Structure diversity of human proto-genes (blue), established human genes (red) and human lncRNAs (orange). The y-axis represents the diversity of the structure. (**d**) Representation of the predicted 2D structure of a randomly selected 5′ UTR of a proto-gene. The green curve is the positional entropy curve, the red curve represents the mountain plot derived from the MFE structure and the black curve represents the pairing probabilities. Asterisks indicate significant differences between the groups (*** *p* < 0.001).

**Figure 8 genes-13-00284-f008:**
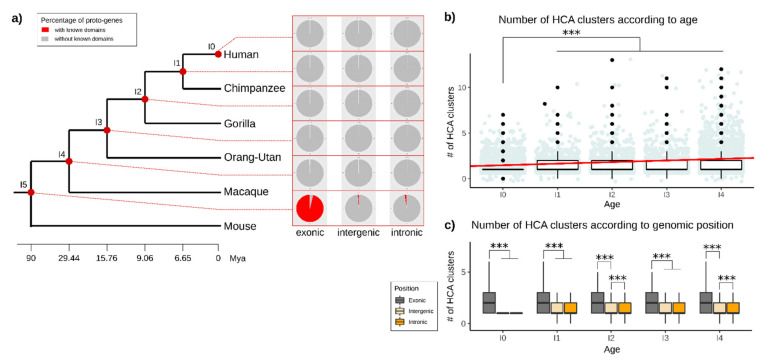
Domains in proto-genes. (**a**) Percentage of detected known motifs in proto-genes, distributed according to the age and genomic position of proto-genes. The red dots linking from the phylogenetic tree to the pie charts indicate the age of the respective results. Each pie chart indicates the percentage of proto-genes with (red) and without (grey) known domains. (**b**) Total number of hydrophobic clusters found in all proto-genes from age class I0 to age class I4. The red line represents the linear model. The y-axis represents the number of hydrophobic clusters found per proto-gene. Blue dots represent the value of HCA cluster for each proto-gene, each new value been represented by a black dot (**c**) Total number of hydrophobic clusters found in proto-proteins according to age (from I0 to I4) and the genomic position. The y-axis represents the number of hydrophobic clusters found per proto-gene. Asterisks indicate significant differences between the groups (*** *p* < 0.001).

**Table 1 genes-13-00284-t001:** Mean proportions of proto-genes with intron.

Age Class	Genomic Position	Mean Proportion of Proto-Genes with Intron
I0	“exonic”	71.4
I1	“exonic”	97.4
I2	“exonic”	84
I3	“exonic”	89.5
I4	“exonic”	90.7
I5	“exonic”	98
I0	“intronic”	11.6
I1	“intronic”	24.25
I2	“intronic”	32.4
I3	“intronic”	31.9
I4	“intronic”	33.95
I5	“intronic”	32.2
I0	“intergenic”	3.2
I1	“intergenic”	5.5
I2	“intergenic”	5.23
I3	“intergenic”	6.22
I4	“intergenic”	8.8
I5	“intergenic”	30.8

**Table 2 genes-13-00284-t002:** Comparison of the number of introns in proto-genes.

Age Class	Genomic Position 1	Genomic Position 2	*p*-Value *t* Test
I0	“exonic”	“intronic”	5.602 × 10^−7^
I0	“exonic”	“intergenic”	5.362 × 10^−8^
I0	“intronic”	“intergenic”	2.2 × 10^−16^
I1	“exonic”	“intronic”	1.972 × 10^−5^
I1	“exonic”	“intergenic”	6.549 × 10^−7^
I1	“intronic”	“intergenic”	1.469 × 10^−15^
I2	“exonic”	“intronic”	2.928 × 10^−5^
I2	“exonic”	“intergenic”	1.269 × 10^−8^
I2	“intronic”	“intergenic”	2.2 × 10^−16^
I3	“exonic”	“intronic”	1.995 × 10^−9^
I3	“exonic”	“intergenic”	2.2 × 10^−16^
I3	“intronic”	“intergenic”	2.2 × 10^−16^
I4	“exonic”	“intronic”	2.2 × 10^−16^
I4	“exonic”	“intergenic”	2.2 × 10^−16^
I4	“intronic”	“intergenic”	2.2 × 10^−16^
I5	“exonic”	“intronic”	2.2 × 10^−16^
I5	“exonic”	“intergenic”	2.2 × 10^−16^
I5	“intronic”	“intergenic”	7.034 × 10^−7^

## Data Availability

The datasets generated and/or analysed during the current study are available in the github https://github.com/AnnaGrBio/De_novoGenes_NewGenomicSignals (accessed on 16 December 2021), and in the supplemental data in https://doi.org/10.5281/zenodo.5786006 (accessed on 16 December 2021).

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
