# Peer review of "New Genomic Signals Underlying the Emergence of Human Proto-Genes"

_genes, 2022, doi:10.3390/genes13020284_

Round 1

Reviewer 1 Report

It is now clear that non-coding ORFs (out-of-frame or intergenic) play an essential role in creating novel genes. Grandchamp et al. present a thorough analysis of the genomic and structural properties of 23135 human proto-genes classified either based on their genomic localization or their "age". The question is timely, and the results are interesting: particularly, the unexpected and novel result on the number of introns which suggests a more complex genomic structure of proto-genes than expected. However, some concepts used in the study deserve to be defined more explicitly to improve the clarity of the manuscript. I also have a few comments that could help the authors strengthen their conclusions. 

***proto-genes/de novo genes/de novo-transcripts

The words proto-genes/de novo genes/de novo-transcripts seem to be used interchangeably in the manuscript (in particular line 32, line 279, lines 305-310...), which is confusing for the reader since it suggests a direct correspondence between de novo-transcripts, proto-genes, and de novo-genes. de novo transcripts are expected (most of the time) to result from pervasive transcription (though some of them may result from regulation) and do not necessarily correspond to genes, even though they can provide the raw material for de novo gene birth. In other words, being transcribed is not sufficient to be a gene. Similarly, proto-genes somehow "precede" genes, offering a reservoir for de novo gene birth, where "expression of non-genic ORFs will occasionally provide an adaptive advantage to the cell" and "adaptive proto-genes will gradually mature under selection, eventually leading to de novo gene birth" (Van Oss & Carvunis 2019). Consequently, these concepts deserve to be defined explicitly, and I suggest the authors either stick to these definitions or explain why they do not, otherwise. 

Also, the authors use the words "proto-genes" and "putative proto-genes". Is there any difference? 

In line with my previous comment, if I have correctly understood, de novo transcripts are novel transcripts, while putative de novo transcribed ORFs are all ORFs contained in de novo transcripts (i.e., there are potentially several de novo transcribed ORFs per de novo transcript). A figure would be highly appreciated to present these different categories. Also, if I am correct, how can an exonic ORF (i.e., resulting from overprinting) be associated with a de novo transcript? (except the case where the borders of transcription are not the same or the spliced product is different, but if I am correct, overprinting is rather related to the translation step). Again, a figure presenting the 3 cases of de novo gene emergence (provided as SI) would be helpful to clarify these definitions. Finally, the authors write: "Each de novo transcript containing a de novo ORF is here described as a proto-gene. ". This sentence suggests that de novo transcripts may have no ORFs (according to ORF definition in ref [33] - i.e., at least 90 nucleotides). What is the fraction of such cases? It seems to be very rare. Also, this sentence reveals that the concept of proto-gene is related to the transcript (i.e., transcript-centered) rather than to the ORF encoding the protein product. Am I correct? In any case, I think it should be mentioned and discussed since (interestingly) these two points of view are different.

***pervasive transcription:

The authors write: lines 104: "The overall transcription is known to be less pervasive in proto-genes compared to older established genes [3]" lines 76-77 "Based on an evolutionary perspective, the older de novo genes become, the more pervasively expressed they are [31], and the more selected ".

Do the authors refer to a more "constitutive" expression for old genes instead of "pervasive" expression? Pervasive transcription (i.e., promiscuous) and constitutive transcription are different things. Older (proto)-genes are rather expected to become more constitutively expressed as proposed in the "out of testis" hypothesis. One should note that the fact that de novo genes are specifically expressed in the brain or testes, or under stress conditions, for instance, does not necessarily indicate "regulation". It rather reflects that in these tissues/conditions, transcription and translation are more "pervasive" (for example, the chromatin in testis cells is in a hypertranscription state).

***UTR analyses:

If I have correctly understood, UTRs are defined according to the de novo transcribed ORF mapped to its corresponding spliced transcript. Depending on the size of the transcript, different theoretical UTRs could be expected. It could be interesting to compare the properties of the detected UTR vs the theoretical ones to see whether the detected one displays different properties and whether these differences are more important in older proto-genes. 

***classes of different ages/genomic position vs HCA: 

The classification of proto-genes according to their "ages"/genomic position is interesting and deserves further analysis. What is the number of ORFs in each class of "ages"/genomic position? What are the corresponding distributions of ORF lengths? 

The authors report an increase in HCA clusters with the ORF age (or among the different genomic positions). It would be interesting to compare this increase with the ORF length of the different age/genomic position categories. Can this increase be explained by the increase in ORF length? Similarly, the authors investigated the GC content of proto-gene UTRs. It would be interesting to compare the GC contents of the different ORF categories (i.e., ages /genomic positions). Indeed, amino acid composition depends on the GC content. Consequently, the number of HCA clusters may be impacted by the GC content of the corresponding ORFs. 

### Other remarks (no particular order):

** While the motivations for searching DNA motifs, introns, or novel domains are clearly presented, those for investigating the "folding potential" of 5'UTR are not so clear and may deserve further explanation.

** lines 143-144: "ORFs with an unusual start or stop codon are only included in the domain annotation, but were removed for the other analyses, giving rise to a file of 23135 putative human de novo transcribed ORFs". I do not understand how START and STOP codons can be predicted without translation data. Does this information come from ref 33? If so, a few words explaining how START and STOP codons were defined would be appreciated. In addition, I do not understand what unusual STOP codons refer to and/or how they were inferred since inferring STOP codons correctly from translation data is difficult due to the low read coverage of these regions. I suppose they are defined according to the last 3' read. In any case, it would be helpful to clarify it.

** lines 214-216: "By comparing the two JASPAR motifs databases with the respective six subsets of DNA sequences using the biopython package [60], we searched for the presence of motifs within the latter. " Not clear what "latter" refers to.

** Figure 2: I do not understand what the "average number of motifs found in the six datasets" refers to? I do not understand how the average has been calculated. I was expected a count of the detected motifs found for each dataset. Do these numbers correspond to the number of occurrences of each motif in each of the six datasets averaged by the number of different motifs? Anyway, it deserves to be clarified.

** line 477: "The longer a sequence is, the less energetic it is in established genes compared to proto-genes" --> the sentence is unclear. It seems to hold both for established and proto-genes.

** lines 497-507: "Unexpectedly, the structure 496 prediction frequency is higher in the 5’ UTR of "intronic" proto-genes (mean = 0.15) and 497 "intergenic" proto-genes (mean = 0.15) than in "exonic" proto-genes (mean = 0.10) (t.test 498 "exonic" vs "intergenic", p-value = 2.2e-16; t.test "exonic" vs "intronic", p-value = 2.2e-16; 499 t.test "intronic" vs "intergenic", p-value = 0.6133). ... We also observed that the probability of 505 the structure was higher in "exonic" proto-genes (mean 62.26) than in "intronic" proto- 506 genes (mean = 54.85) and "intergenic" proto-genes (mean = 55.84) (t.test "exonic" vs 507 "intergenic", p-value = 3.181e-09; t.test "exonic" vs "intronic", p-value = 5.074e-12; t.test 508 "intronic" vs "intergenic", p-value = 0.4401). "

These two results seem contradictory and deserve to be commented on. 

** Figure 7, the legend of the last panel is missing. Also, I suggest the authors replace "Number of HCA domains" with "Number of HCA clusters" since the use of "domain" is confusing (domain is a dedicated word).

Reviewer 2 Report

The manuscript by Grandchamp et al is an interesting paper. It provides good evidence for a progressive development of a number of protein parameters in proto-genes with evolutionary time. These parameters are: transcriptional promoters (called motifs), introns, 5’UTRs, and protein domains. The results add another layer to our understanding of how genes evolved from proto-genes. I have several comments and questions that are meant to improve the paper.

    1. Throughout manuscript I would change ncRNA (a broad term that includes miRNAs) to lncRNA.

  1. p.2, line 84, some terminology used is not common. For example, in the Introduction: clearly define “overprinting”, “exonisation”, or “from scratch”. Also on terminology, I assume overprinting is the same as transcriptional readthrough. But would you consider overprinting of an existing gene with addition of a new functional sequence to be in a proto-gene genomic region? I would have a hard time saying yes.

  1. p.13 line 455. It would be of interest if in future investigations there was more about the 3’ UTR properties of proto-genes relative to established genes. The 3’ UTRs have such interesting and varied functions [Mayr C. Cold Spring Harb Perspect Biol. 2019 Oct 1;11(10):a034728], much involving the stem loop preceding the terminal AAAA… that functions in protein binding. The development of the secondary structure of the stem loop from proto-gene to an established gene would be interesting.

  1. p.14, line 47. Clarify terminology: “5' UTR of non-coding RNAs.” Do you mean "pseudo 5' UTR" of some lncRNAs that bind ribosomes (see Statello L, et al. Gene regulation by long non-coding RNAs and its biological functions. Nat Rev Mol Cell Biol. 2021. PMID: 33353982)? Obviously ncRNAs do not have 5’ UTRs. Line 474, again, clarify in manuscript the use of ncRNA.

  1. p.15, lines 493,494 you would expect proto-genes or established gene 5' UTRs to be highly structured as they function in ribosome binding but ncRNAs for the most part do not bind ribosomes, so the ncRNA should perhaps be considered as a control.

  1. p.15, Fig. 6a, right. The 5’ UTRs of mRNAs are approximately 50-200 nt in length. Thus the 1000 bp length shown in Fig. 6 seems unrealistic. I would suggest adding a duplicate and magnified drawing in the form of an insert box showing the length only up to 250 bp. This could be added in the bottom left of the figure. I believe that still shows a big difference between proto-gene and established gene in energy gene 2nd structure. Also define better energy of 2nd structure or alter terminology, such as minimum free energy.

    7. Figure 6d. It would help the reader if you indicate the 5' end of the secondary structure with an arrow so it does not look like a closed circle. Or use the PGN or GIF formats so one could better see the nucleotide residues (I assume you are using

     [http://rna.tbi.univie.ac.at/cgi-bin/RNAWebSuite/RNAfold.cgi]. The structure shown in Figure 6d is very interesting. It looks rather stable with a considerable # of GC pairs in hairpins. Striking also is that it resembles the backbone of a pseudoknot. Many mRNA 5' UTRs have pseudoknots. This structure may suggest the proto-gene structure is on its way of developing a pseudoknot??

    8. p.9, line 332, typo “transcripts which ORF was”. There are other typos.
